# Metaverse and Sustainability: Systematic Review of Scientific Publications until 2022 and Beyond

Jussi S. Jauhiainen [1,2,*], Claudia Krohn [1] and Johanna Junnila [1]

1   Department of Geography and Geology, University of Turku, 20014 Turku, Finland
2   Institute of Ecology and the Earth Sciences, University of Tartu, 51014 Tartu, Estonia
*   Correspondence: jusaja@utu.fi

**Abstract:** The metaverse is the next evolution of the Internet, merging the physical and digital worlds into a multiuser environment. It is discussed widely in the media and among technology developers. It may expand to many aspects of society and people's everyday lives. In this article, we examine how academic discussion and research about the metaverse developed from the 1990s to the end of 2022. We focus on the quantitative development of scientific publications about the metaverse, the key countries and organizations behind these publications, the key research topics and areas, and whether and how those publications addressed sustainability. We identified 491 international scientific publications (peer reviewed articles, reviews, and proceedings papers) related to the metaverse in the Web of Science database and 2240 scientific publications in the Scopus database, between 1995 and 2022. The number of publications is rising very fast as most of publications on the metaverse were published in 2022. Scholars in universities and research institutes in the United States, China, the United Kingdom, and South Korea are the most frequent publishers. Publications very seldomly address sustainability as the main subject. Usually, sustainability is considered very narrowly, despite the metaverse's large and significant expected future economic and social impact. Sustainability and responsibility should be integrated into the design, construction, and development of the metaverse and related research.

**Keywords:** metaverse; sustainability; virtual reality; extended reality; scientific publication; Meta; China; the United States; digitalization; Web of Science; Scopus

## 1. Introduction

The metaverse is the next evolution of the Internet, merging the physical and digital worlds into a multiuser environment. It is meant to enable an immersive real-time presence and interaction in multisensory interactions with virtual environments, digital objects, and people in digital, often in three-dimensional (3D) spaces [1,2]. The metaverse is expected to expand to many aspects of society and people's everyday lives in work and free-time environments. Instead of us being on the Internet, we will be "in" the Internet [3,4]. On the one hand, the metaverse concerns the development of hardware and digital infrastructure, including related devices and support services. On the other hand, it includes various kinds of software that facilitate its use and functional activities, including shopping, gaming, communication, and other types of social interaction. It also suggests major changes in the monetary system, with the emergence of digital currencies and assets, as well as nonfungible tokens (NFTs), and includes the creation of digital twins, regarding the material environment and people (Figure 1).

Park (2022) presented the widely used categorization of the metaverse elements into six realms [6]. These are physical devices and sensors, recognitions, rendering, and technical methods that all refer to technical aspects of the metaverse; user interaction and scenario generation, indicating how communication and interaction are organized in the metaverse; and applications that refer to concrete realms in which the metaverse is applied, such as

work (office), learning (education, schools), entertainment (games), social communication and networking (social), consumption (marketing), and digital twins and their activities (simulation) (Figure 2).

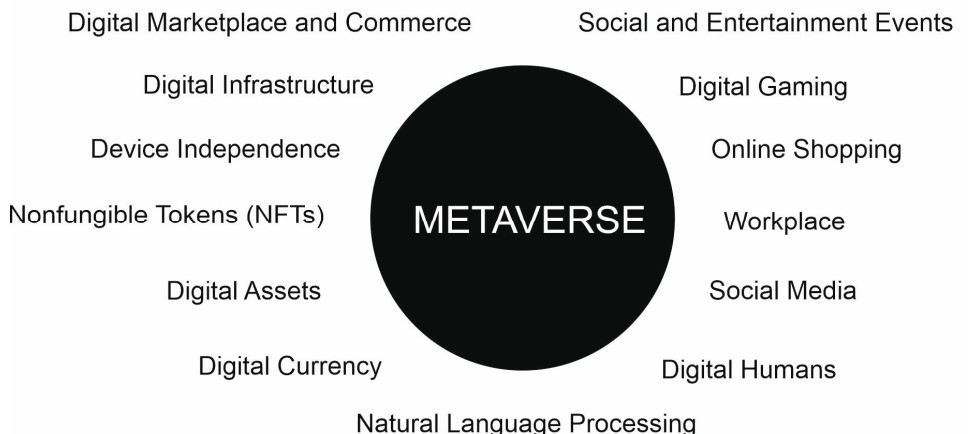

**Figure 1.** Metaverse. Source: Modified from [5].

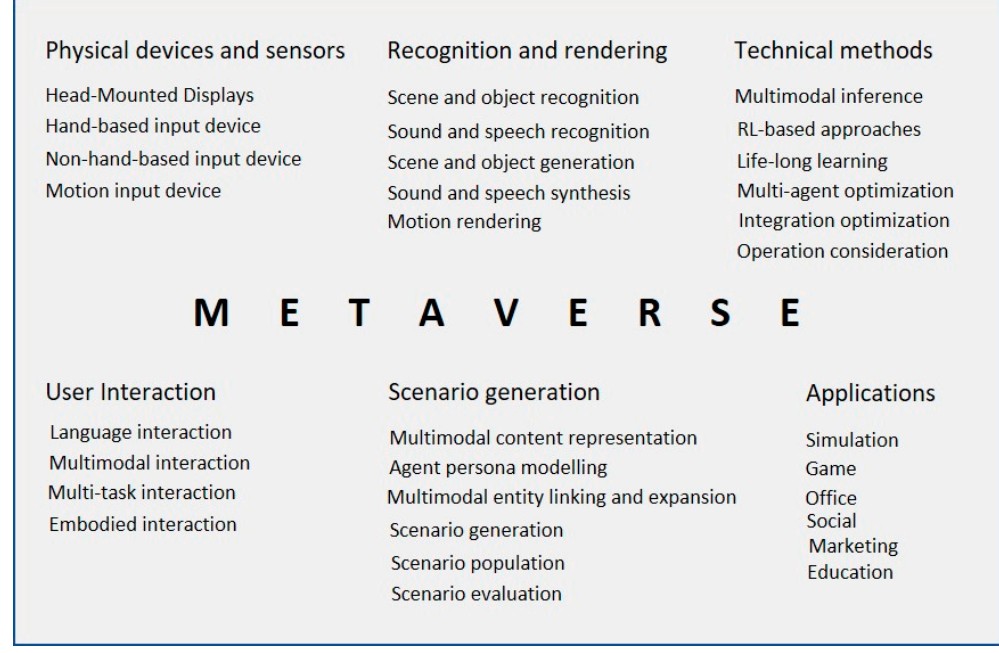

**Figure 2.** Dimensions of the metaverse. Source: Modified from [6].

The prerequisite for the metaverse is that digital technology, communication networks, and applications develop quickly and become globally ubiquitous and easy and affordable to use, resulting in a multitude of users who can access virtual worlds, digital objects, and each other in the 3D Internet. For a basic, less immersive experience of the metaverse, one needs access to the Internet; suitable applications; and a mobile phone, tablet or laptop. For a more immersive experience, one needs more specific devices and interactions. In virtual reality (VR), the simulated environment and world are experienced through digital technological items, such as VR headsets, whereas in augmented reality (AR), the physical world's material environment is augmented with digital 3D elements. In a more intensive and immersive experience of extended reality (XR), the physical environment, people, VR, and AR blend together, creating added value. The metaverse mostly entails the latter immersive practice [1,2].

Technological development is necessary for the metaverse. However, the interest in its development has arisen due to its expected large social and economic impact. The metaverse may bring billions of people into a globally shared interactive digital environment, as various social media platforms have done [7]. This is also seen in the development of related businesses. For example, the owner of the widely used social media platform Facebook changed the company's name from Facebook to Meta [8]. According to recent estimations, the metaverse, as a materially and digitally expanded XR environment, will contribute to major changes in current societies and their economic infrastructure, industries, and employment. Several early estimations indicate that the metaverse will increase the global GDP (gross domestic product) by several percentage points in the 2030s and be worth trillions of dollars [2,9,10]. However, no one can know yet whether there will be a single dominating metaverse or many metaverses with small digital environments. The metaverse is still in an emergent phase, so not all visions of it coincide.

Remarkable amounts of money and other resources are currently invested in metaverse development. With the climate crisis, biodiversity loss, and global inequality as some of the most pressing issues of the time, sustainability is also a key topic for the metaverse's responsible future, and it is very relevant to study how sustainability is taken into account in current research relating to the metaverse ([11–16]; see Section 2). For this, one needs to scrutinize the metaverse-topic research conducted so far. As evidenced later in this article, though the number of reviews regarding the metaverse publications is on the increase, and the number of scientific publications about the metaverse is growing very fast, there has not yet been any scientific publications that have focused on sustainability and the metaverse.

A fully immersive and visually rich real-time metaverse is still in the formative stage and years away from completion. Therefore, much discussion about the metaverse is speculation of its potential future—what the metaverse will be, when it will arise, and what will be its impact [10]. Nevertheless, research and academic discussion on the metaverse are developing quickly. According to our study, in the Web of Science (WoS) and the Scopus databases, more than twice as many metaverse-related peer-reviewed scientific publications appeared in international journals in 2022 than had ever been published. The growth of publications in other metaverse-related research and other metaverse publications is even faster. VR and AR technologies are advancing, and the demand for immersive Internet is increasing, so related research and academic scholarship will develop quantitatively and qualitatively. Research and the number of publications on the metaverse are expected to expand rapidly in the coming years.

In this article, we examine research and academic discussion about how the metaverse emerged and developed in scientific publications, in international journals and its state-of-the-art at the end of 2022. We asked the following research questions: (1) How did the number of scientific publications about the metaverse increase until the end of 2022, and what were the key countries and organizations behind these publications, (2) What were these publications' key research topics and areas, and (3) How were sustainability and the metaverse connected in the publications if at all?

After the introduction, we present the conceptual framework of sustainability for the metaverse's material and digital realms. For this purpose, we use the United Nations Sustainable Development Goals (SDGs) as a broad reference. Then we discuss material and methods for the article, i.e., the publications in the WoS and the Scopus databases with the topic of the metaverse, following the PRISMA (Preferred Reporting Items for Systematic Reviews and Meta-Analyses) methodological approach. We then present the results, focusing on the quantitative development of publications on the metaverse, their publication venues and research areas, geographical and organizational backgrounds, the publications' content, and how they addressed sustainability. Finally, the article ends with a discussion, answering the research questions and suggesting key research avenues regarding the metaverse.

## 2. Metaverse(s) and Sustainability

The metaverse and sustainability are connected in many ways. In fact, sustainability is a fundamental issue because it is hypothesized that the metaverse would significantly affect the material (environmental), economic, and social spheres, as well as sustainability around the world [11]. The result of increasing consumption and population growth is that many crucial natural resources will end if they are not used sustainably, and their recycling organized accordingly. The media and many enterprises discuss that while the metaverse would boost economy and financial profitability, these need to be put in balance with environmental, social, and ethical aspects of the metaverse [13].

On the one hand, the metaverse may result in substantial reductions in carbon emissions as many physical goods will be substituted with digital ones, physical mobility and construction are reduced with virtual interactions, the physical world is optimized with digital twins, simulations can be performed in cyberspace instead of material space, and larger awareness and more efficient interaction are created against harmful global phenomena, such as climate change. The emerging digital spaces may cause less environmental harm than their predecessors due to more advanced technology. Radically, a more dematerialized future is within reach, if people intend to buy fewer physical items, because more things can be purchased and consumed only digitally [14,15].

Many technology development companies prioritize sustainability goals toward more energy-efficient practices that shrink their current carbon emissions [16]. For example, the metaverse can support green networking technologies including minimizing the number of physical data centers, by storing data and information in the cloud [17]. It is important to design the metaverse platforms so that fewer specialized hardware resources and high computing power are required. It might be more efficient to have a small number of metaverses instead of thousands of metaverses developed by a very large number of organizations [12].

On the other hand, the design, construction, and use of technology devices and wider and faster physical data networks require energy and other natural resources, and produce $CO_2$ emissions. Cryptocurrencies and NFTs rely on blockchain technology, which uses a significant amount of energy [11]. Additionally, cloud computing data centers require large amounts of energy, primarily used to cool the heat generated by the servers. For immersive experiences, new hardware devices are needed, thus requiring the use of rare earth metals and at the same time creating more electronic waste, when old devices are abandoned. However, within the blockchain space, a major shift toward sustainability is emerging. The energy heavy 'proof of work' principle in digital mining, in the creation of NFTs to be used in the metaverse, is changing towards less energy-intensive 'proof of stake' chains [11,14,15].

At the same time, one needs to address the social sustainability. In the metaverse, many kinds of data are collected from its users. It can be an unethical place if guidelines and regulations in data collection and uses are not designed and implemented properly. Otherwise, the metaverse may lead into an erosion of trust. That would compromise its enormous potential as a complement to physical face-to-face transactions [15].

As evidenced above, the metaverse needs to be connected and analyzed along broad realms of sustainability. One possibility is to connect the metaverse with the United Nations' SDGs (Figure 3, Table 1), which have been accepted as global goals to be achieved locally around the world. All United Nations member states adopted these 17 goals in 2015. They are not only goals but also an urgent call for action in global partnership. Ending poverty and other deprivations requires strategies and actions that improve health and education, reduce inequality, and spur economic growth, and these efforts are necessary to address climate change and preserve life on land and in water [18].

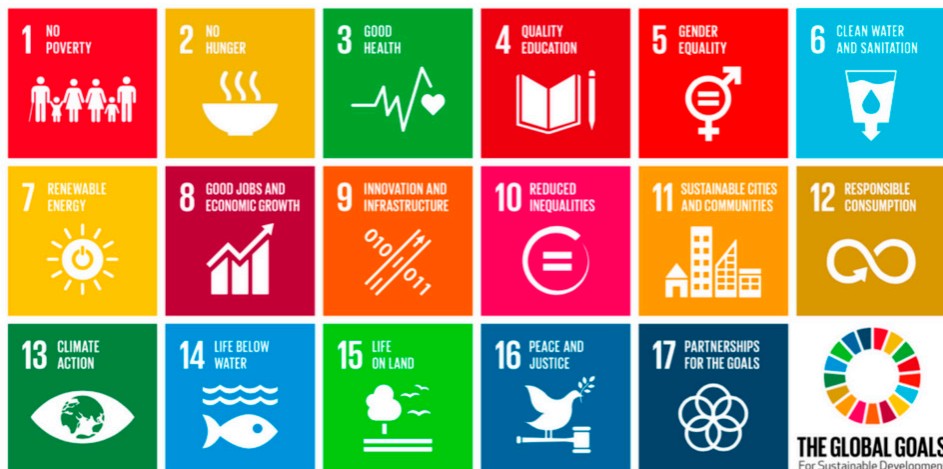

**Figure 3.** Dimensions of sustainability: SDGs. Source: [18].

**Table 1.** SDG dimensions of the metaverse's digital and material realms.

| | Metaverse as a Digital Realm | | Metaverse as a Material Realm | |
|---|---|---|---|---|
| **SDG Nr** | **Opportunity** | **Challenge** | **Opportunity** | **Challenge** |
| 1 No poverty | No poverty | Not addressed | Reduction | Growth of poverty |
| 2 No hunger | No hunger | Not addressed | Reduction | Growth of hunger |
| 3 Good health | Awareness | Not addressed | Better health | Decreased health |
| 4 Quality education | Improvement | No access | Wide access | Selected access |
| 5 Gender equality | Wider equality | Inequality | Wide access | Selected access |
| 6 Clean water and sanitation | Not a problem | Not addressed | Less use | Pollution |
| 7 Renewable energy | Awareness | Not interested | Less energy use | Increased energy use |
| 8 Good jobs and economic growth | Wider access | Exploitation | Expansion | Polarization |
| 9 Innovation and infrastructure | Promotion | Inequality | Technology | Unavailability |
| 10 Reduced inequalities | No inequality | Sharp inequality | Promoting equality | Rising inequality |
| 11 Sustainable cities | Smart city | Divided city | Smart city | Divided consumption |
| 12 Responsible consumption | Modesty | Expansion | Less consumption | Increased consumption |
| 13 Climate action | Awareness | Not addressed | Less emissions | More emissions |
| 14 Life below water | Awareness | Not addressed | No impact | More consumption |
| 15 Life on land | Inexpensive | Expensive | Less density | Inequal access |
| 16 Peace and justice | Collaboration | Digital wars | Collaboration | Increased hostilities |
| 17 Partnerships | Global | Selected teams | Network | Partial accessibility |

The metaverse as a global platform will become one important arena in which SDGs can be addressed. The metaverse could play an important role in the achievement of various SDGs, not only addressing them in the digital realm, but also by transforming the material word connected to SDGs. However, the metaverse's development, construction, and implementation can create challenges to sustainability. Therefore, the metaverse could be scrutinized, first as a digital twin of the material world, and to see how various aspects of such a digital (immersive) space would affect environmental, economic, and social sustainability (Table 1).

The second aspect, intertwined with the first one, is the metaverse's effects on sustainability in the material world, i.e., what environmental, social, and economic sustainability dimensions are involved in the metaverse's development, design, and implementation (Table 1). Although the metaverse is a digital 3D Internet realm, it is based on material infrastructure, and requires the use of natural resources and energy for its production and use, as discussed above.

### 3. Materials and Methods

To study the scientific publications on the metaverse quantitatively, there are two major suitable databases. The first is the Web of Science database. This database, by Clarivate, owned by Thomson Reuters, contains more than 18,000 scientific journals and tens of millions of publications from many disciplines. Other databases also contain scientific publications. For example, Google Scholar and the Scopus databases contain more books, journals, and articles than the WoS. However, the publications contained in the WoS usually have greater scientific impact than those outside of it [19]. Furthermore, if one focuses only on peer-reviewed international scientific articles in international journals, as we do in this article, the WoS is a very suitable database for that purpose. Another database used in this article is the Scopus abstract and citation database that is owned by Elsevier, another major international publisher. It contains almost 26,000 peer reviewed journals, however, not all of them have an international editorial board.

We studied peer-reviewed scientific publications about the metaverse in international journals, from the earliest publications referring to it from the 1990s to the most recent ones in 2022. The scientific publications selected for this study in the WoS database appeared in journals and proceedings that all implemented rigorous peer reviews, had an international editorial board, published several issues annually, and were in English, and the scientific publications in the Scopus database followed a rigorous peer review process, were in English, and appeared in active publication venues.

For the WoS, we did not include the empirical sample books, book chapters, chapters in seminar publications or letters because they were not necessarily peer reviewed, thus failing the scientific scrutiny and review, before their publication. This was to avoid anomalies in the sample. For example, due to the small number of metaverse-related scientific publications before 2022, a single non-refereed edited book with chapters on the metaverse would have generated more publications than all peer-reviewed metaverse-related publications that year. Furthermore, opinion papers, letters, and seminar publications would not necessarily provide more peer-reviewed metaverse-related publications, not to mention hundreds of articles in the media. Our selection is not to claim that other publications would not have value regarding the studies about the metaverse, but they do not necessarily fulfill the criteria of scientific peer-reviewed publications and research, which is the scope of this article.

The material for this study consists of two databases. The first database regards peer-reviewed scientific articles, scientific review articles, and scientific papers in proceedings that had the metaverse as their topic in journals, that are indexed in the WoS database, and that appeared in English. The topic (the WoS category "Topic") was the metaverse when the scientific publication's title, keyword, summary or the full text mentioned the word "metaverse" or "metaverses", and the publication was about the metaverse. The first such article was from 1995, and the newest articles appeared in December 2022. The second database regards peer-reviewed scientific publications that had the metaverse as their topic in journals, that are indexed in the Scopus database, and that appeared in English.

Screening the entire WoS database (as of 1 December 2022) resulted in three datasets, after removing doubles and articles that on a closer look turned out to be about something entirely different than metaverse (e.g., a medical one about metaversion as a physiological concept): (1) all metaverse articles until the end of 2021 (158 articles), (2) all metaverse articles appearing in 2022 (333 articles), (3) and all metaverse articles until December 2022 (491 articles), i.e., the two previously mentioned datasets together (Figure 4).

Screening the Scopus database (as of 19 December 2022) resulted in three datasets: (1) all metaverse articles until the end of 2021 (1080 articles), (2) all metaverse articles appearing in 2022 (1160 articles), and (3) all metaverse articles until December 2022 (2240 articles), i.e., the two previously mentioned datasets together.

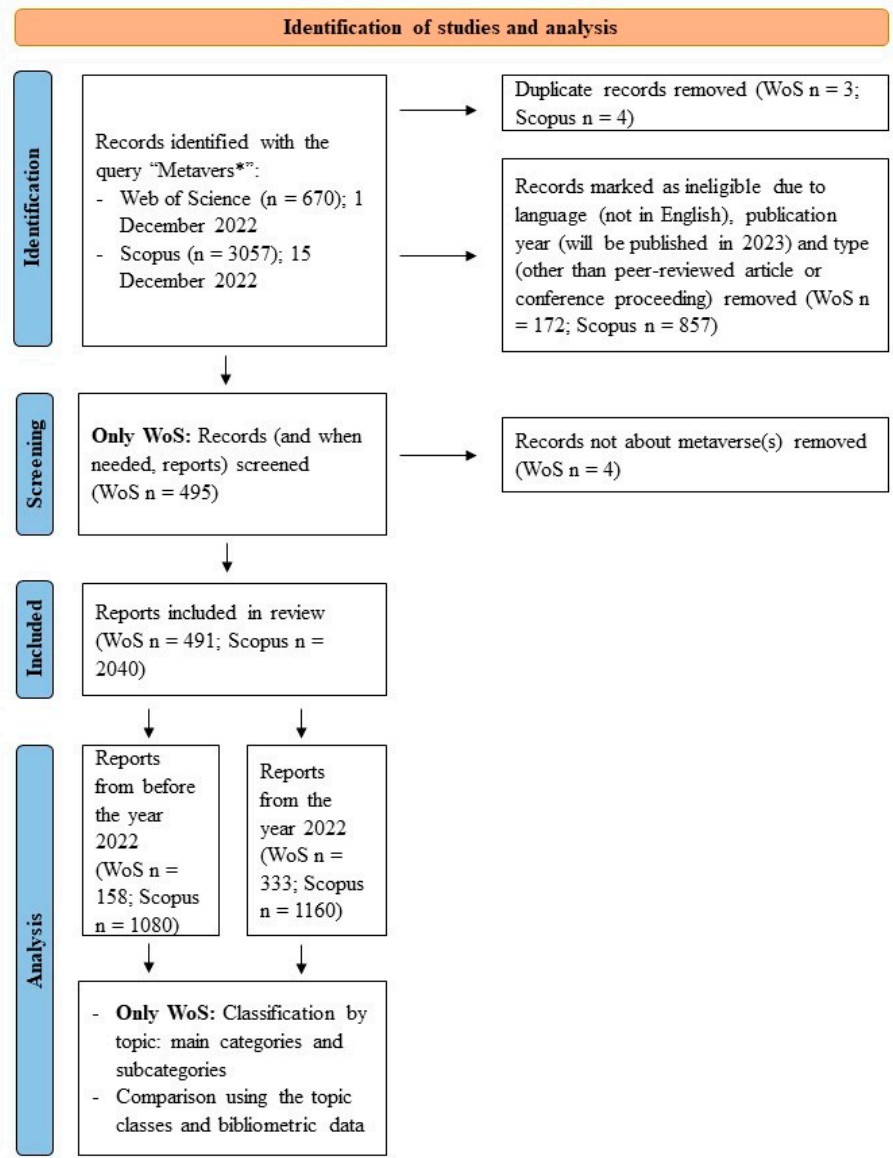

**Figure 4.** Flow chart of the data gathering and analysis of the WoS and Scopus databases with the search command "Metavers*". Template modified from [20].

First, we analyzed the volume of metaverse-related publications until 2022 and in 2022, both in the WoS and Scopus databases.

Second, we determined the most common research topics, themes, countries, and organizations in these publications, in the WoS and Scopus databases. For the more detailed analysis, we used the VOSviewer program (visualization of similarities) and its algorithms to analyze the WoS, to find connections among the topics and research areas (using author keywords) and organizations (using author affiliation countries). It is a commonly used technique for analyzing publications in the WoS (for details, see, e.g., [21]).

Third, we conducted content analysis on the metaverse-related publications in the WoS to identify their main and secondary topics or point of views to the metaverse, using five classes (technology, uses, societal influences, sustainability, and multi-topic reviews) and 18 subclasses. The classification was conducted based on the abstracts and metadata, with the help of the full text version, in unclear cases if it was available. We quantitatively analyzed the materials mainly using descriptive statistics, focusing on the differences in these datasets.

In the careful collection and analysis of the material, research ethics were followed closely, indicating also the financial publishing corporations behind the databases used, namely Thomson Reuters (WoS) and Elsevier (Scopus). However, both organizations claim to follow the practice in which they do not modify the databases to promote the publications from journals they own, and that the databases are curated by independent subject matter experts.

## 4. Results

### 4.1. Number of Publications

According to the WoS and Scopus databases, the first scientific article having the topic "Metaverse" appeared in 1995, more than a quarter of a century ago [22]. This article addressed the situation in which one could move from one virtual world to another. However, at that time, digital technology had not advanced yet to the levels to make it possible. Therefore, the article was merely a future speculation of such virtual worlds.

The initial article did not generate much wider interest in the topic. In the next eight years, from 1996 to 2003, only two scientific articles were published on the topic in the WoS and eight in Scopus. From 2004, at least one scientific peer-reviewed article or conference paper was published annually. In the WoS, the number of publications per year remained less than ten, until 2020, with the exception of the years 2009–2013, when 11 to 19 papers fulfilling the criteria were published annually. Likewise, in Scopus, a larger number of publications appeared during the same period (80–106 annually). This "peak" regarded the so-called proto-metaverses, such as The World of Warcraft, Roblox, and Second Life, and their virtual, game-oriented popular culture. The metaverse was generally considered a virtual game environment that could reach players globally. In particular, Second Life attracted the attention of scholars. It was a multimedia platform in which people could make an avatar for themselves and interact with other users, in a multi-player online virtual world of which contents were created by the users. In addition, many articles mentioned the sci-fi cyberpunk book, *Snow Crash*, by Neal Stephenson, which was published in 1992. In the book, humans were programmable avatars in a 3D virtual environment [23]. During the 2010s, the metaverse started to be mentioned as a new technology for applications in society, such as digitalization of businesses and e-learning in education. Furthermore, the articles of the time discussed related technological aspects mostly separately, such as VR and AR, in the formation of the metaverse-related digital environment. Among the more recent publications in the 2020s, Second Life was no longer the publications' main topic. Sometimes it is still referred to as an early example of a metaverse.

The publication numbers changed substantially in 2021. In the WoS, a fivefold growth took place from the previous year (32 publications appeared), and in 2022 such growth was over tenfold (333 publications appeared) (Figure 4). In Scopus, the number of publications doubled from 2020 to 2021 (117 publications appeared), and tenfold from 2021 to 2022 (1160 publications appeared). This growth also brought about a conceptual change in the academic discussion. Instead of only discussing a separate virtual world with a focus on technological devices, researchers paid more attention to the metaverse as an immersive platform. Various technologies co-existed, forming the metaverse, in which physical and virtual realities were combined. A specific event in October 2021 led to a worldwide increase in the media representation of the metaverse, namely the formation of Meta Platforms, Inc. (Menlo Park, CA, USA) [6]. The number of articles containing the word "metaverse" started to increase rapidly. Mark Zuckerberg's vision of transforming Facebook's social media network to the metaverse substantially increased scholars' attention. Since the autumn of 2021, positive and critical references have been made to this matter. The number of peer-reviewed scientific articles, scientific review articles, and scientific papers in proceedings on the metaverse topic is expected to surpass 1000 in the WoS database in 2023, and become several thousand in the Scopus database in 2023.

*4.2. Publication Venues and Research Areas*

Until the year 2021, there were 158 peer reviewed scientific publications in the WoS containing the topic "metaverse". Of these, half (49.4%; 78 publications) were proceeding papers and the other half (49.3%; 77 publications) were scientific articles. Very few (1.9%; 3 publications) peer-reviewed publications were reviews, as until then, only a rather small number of reference material for that purpose had appeared. The largest share of publications being conference proceedings indicated the emerging character of the topic. In Scopus, until the year 2021, there were 1080 scientific publications containing the topic "metaverse". Of these, less than half (41.5%; 448 publications) were conference papers and more than half (54.5%; 589 publications) were scientific articles. Few publications (4.0%; 43 publications) were reviews, but were absolutely and relatively more than in the WoS.

This publishing situation changed in 2022. In the WoS, 333 peer reviewed scientific publications with the topic of the metaverse appeared, i.e., more than doubling the scientific scholarship on the metaverse so far. Substantially more, i.e., over two out of three publications were articles (70.9%; 236 publications), and a substantially lower share than before, about one out of five (20.1%; 67 publications) were proceedings papers. The number and share of reviews over the metaverse had grown rapidly (14.7%; 49 publications). Furthermore, in Scopus, a rapid growth of metaverse-related publications took place. Out of 1160 publications, 62.4% (724 publications) were articles, 28.4% (329 publications) were conference papers, and 9.2% (107 publications) were reviews.

In the WoS, between 1995–2022 there were published peer reviewed scientific publications on 348 different journals or conference proceedings. The most common ones were *2022 IEEE Conference on Virtual Reality* and *3D User Interfaces* (14 articles), *IEEE Access* (14), *Sensors* (13), *Sustainability* (12), and *Electronics* (9). In Scopus, in 2022, the most common venues were *Sustainability* (35 publications), *ACM International Conference Proceedings Series* (23), *IEEE Access* (22), *Electronics* (22), and *Frontiers in Psychology* (20).

Overall, of all these publications in the WoS, almost half (48.7%) were published through the Open Access modes so that these would be reachable by potential readers through the Internet, without needing to pay for them or having to be affiliated with organizations that would have access to the international journals, in which they appeared. In this aspect, a substantial change occurred over the years. Of the metaverse-topic publications before 2022, only slightly more than one out of four (27.2%) were available through Open Access. In 2022, the majority (58.9%) of publications followed the Open Access practice, facilitating the immediate information flow to the readers. This made it easier to follow the state-of-the-art development in the scholarly discussion this year, compared to a few years before. In Scopus, of all metaverse-related publications, 38.6% were published through the Open Access modes; 24.8% of publications before 2022 and 51.4% in 2022. Open Access was thus more common in the international scientific peer-reviewed publications about the metaverse, indexed in the WoS than among the wider scope of metaverse-related publications in the Scopus.

Scientific research fields on metaverse topics focus extensively on computer science and engineering, both in the WoS and the Scopus. They most often involve human-computer interaction. In the WoS, other less frequent but rapidly evolving fields were business, economics, and telecommunications. Management and communications were the most common subfields. Before 2022, there were relatively more publications on educational research, but in 2022, its proportional share diminished. Instead, in other science and technology topics and specific science fields (such as materials science and physics), researchers started paying more attention to the metaverse, as the rapidly growing number of scientific publications indicated. However, in 2022, more in-depth conceptual-ontological views on the metaverse also emerged, because they were also discussed, for example, in literary theory articles. In Scopus, there were more social scientific publications about the metaverse compared with those in the WoS (Table 2).

**Table 2.** Research fields in scientific publications about the metaverse in the WoS and Scopus database categories.

| Research Field | Number of Publications | | | | | |
| --- | --- | --- | --- | --- | --- | --- |
| | WoS | Scopus | WoS | Scopus | WoS | Scopus |
| | 1995–2021 | | 2022 | | Total | |
| Computer science | 80 | 706 | 134 | 668 | 214 | 1384 |
| Engineering and material science | 51 | 736 | 124 | 772 | 175 | 1518 |
| Health, medical and physiologicalsciences, psychology | 3 | 102 | 167 | 203 | 170 | 305 |
| Other natural sciences and technology | 23 | 123 | 111 | 354 | 134 | 480 |
| Business and economics, management | 23 | 177 | 34 | 157 | 57 | 334 |
| Political and social sciences, law | 9 | 345 | 32 | 268 | 41 | 614 |
| Education and educational research | 17 | .. | 14 | .. | 31 | .. |
| Art, humanities and linguistics | 15 | 102 | 9 | 84 | 24 | 186 |
| Environmental science and ecology | 4 | 23 | 17 | 69 | 21 | 92 |

### 4.3. Geographical Origin and Networks

Before the year 2022, scientific publications on the metaverse in the WoS involved authors and organizations from 28 countries, of which 15 countries published three or more publications. The most common countries to publish were the United States (20.2%, 32 publications), South Korea (15.2%, 24), the United Kingdom (7.6%, 12), Spain (7.6%, 12), and Japan (7.0%, 11). Of the 27 European Union member states, 13 were involved in these publications. None of the publications were from Africa, and nine (6.5%) were from South America. The geographical distribution was proportionally broad: the three main countries represented clearly less than half (43.0%) of all publications. Very few publications linked authors and their organizations across countries, so articles were mostly isolated from each other. In Scopus, metaverse-related publications before 2022 appeared from scholars in institutions in far more countries compared to the WoS, namely from 77 countries. Publications were most common from the United States (27.6%, 298 publications), the United Kingdom (11.9%, 128), Australia (6.8%, 73), China (5.3%, 62), and South Korea (5.4%, 58).

In 2022, with the rapid growth of the publication numbers, changes occurred in the geographical origin and networks behind publications. China quickly became the most common country involved in the metaverse related publications both in the WoS and Scopus. In the WoS, China (31.1%, 103) was followed by South Korea (18.0%, 60), the United States (14.4%, 48), the United Kingdom (10.8%, 36), and Germany (5.7%, 19). Overall, 54 countries were involved in publications, doubling the number of countries thus far, and of these, 37 countries published three or more publications. Of the 27 European Union member states, 19 were involved in these publications. Geographical clusters of the metaverse-topic research emerged and networks between countries became denser (Figure 5). The overall geographical origin of all metaverse-topic scientific publications in 2022 showed a substantial concentration of publications in a few countries, as almost two out of three (63.4%) publications came from the top three countries. Very few African, Central Asian, and Latin American countries were involved in the academic scientific research and discussion on the metaverse. African countries featured in 11 (3.3%) of the publications and even fewer (1.8%, 6) were from South America (Figure 6). In Scopus, the most common countries from where the metaverse-related publications emerged in 2022 were China (23.5%, 273 publications), the United States (16.7%, 194), the United Kingdom (10.9%, 126), South Korea (10.2%, 118), and India (4.8%, 56) (Figure 6).

From 1995 to 2022, in the WoS, more than 300 organizations, usually universities and research institutes, were involved in scientific publications about the metaverse. Of those organizations, 172 contained at least two publications about the metaverse, 20 had at least five, and three had at least 10, namely the Sepong University in South Korea (19 publications), the Chinese Academy of Sciences in China (12 publications), and the Sabanci University in

Turkey (10 publications). In Scopus, the most common organizations publishing about the metaverse were the Chinese Academy of Sciences in China (26 publications), the Nanyang Technological University in Singapore (20 publications), the National University of Singapore in Singapore (20 publications), the Suzuka College of the National Institute of Technology in Japan (19 publications), the University of Kentucky in the United States (18 publications), and the Nagaoka University of Technology in Japan (18 publications).

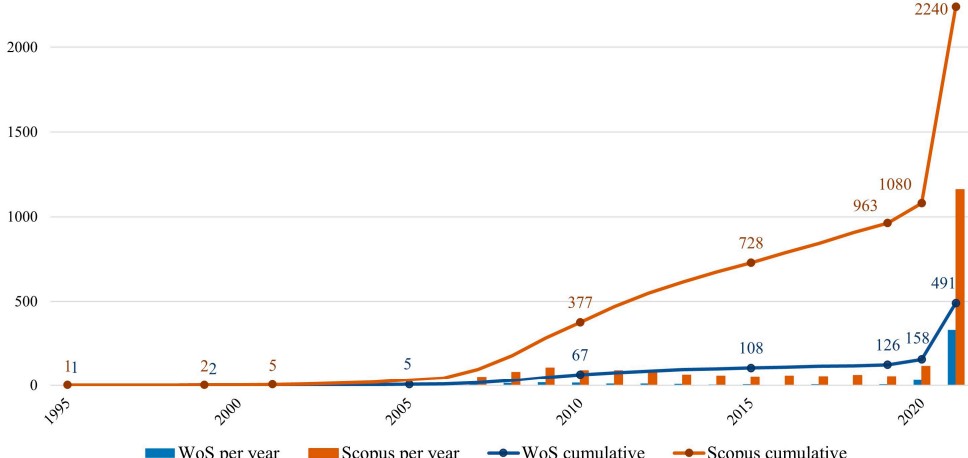

**Figure 5.** Number of metaverse-topic scientific publications in the WoS and Scopus databases, 1995–2022. Calculated from the WoS and Scopus databases.

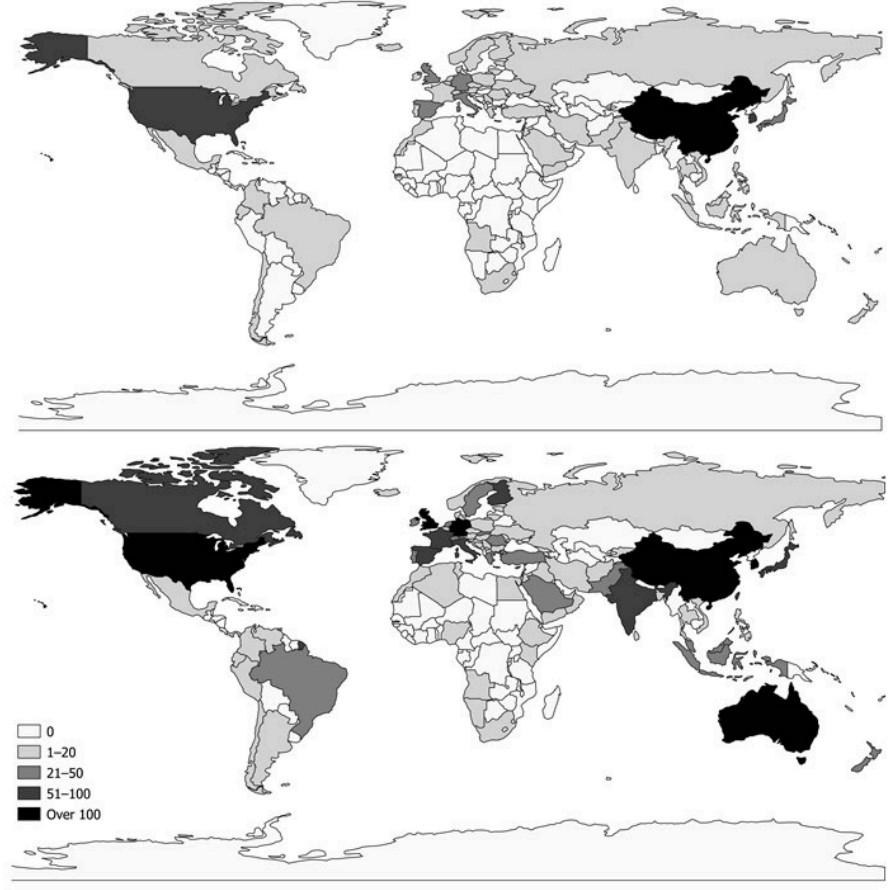

**Figure 6.** Geographical origin of scientific metaverse-topic publications in the WoS (**top**) and Scopus (**bottom**) databases, 1995–2022. Calculated from the WoS and Scopus databases.

### 4.4. Contents of Publications

For the specific analysis of the contents of publications, we focused the analysis only on the WoS database. This was to delimit the analysis only on scientific publications that had been published in peer reviewed international journals and conference proceedings. This indicated the contents of the core scientific research about the metaverse. Between 1995 and 2021, the authors of all scientific metaverse publications mentioned in the WoS 582 keywords, and 29 of them were mentioned at least three times (Figure 7). The five most commonly mentioned self-defined keywords were "metaverse[s]" (63 times, in 40.5% of publications), "virtual world[s]" (38, 24.1%), "Second Life" (29, 18.4%), "avatar" (13, 8.2%), and "virtual reality" (11, 7.0%). In 2022, the five most common self-defined keywords among all 1212 mentioned keywords were "metaverse[s]" (170 times, 53.8% of publications), "virtual reality" or "vr" (84, 25.2%), "augmented reality" or "ar" (38, 11.3%), "blockchain[s]" (27, 8.1%), and "digital twin[s]" (19, 5.4%).

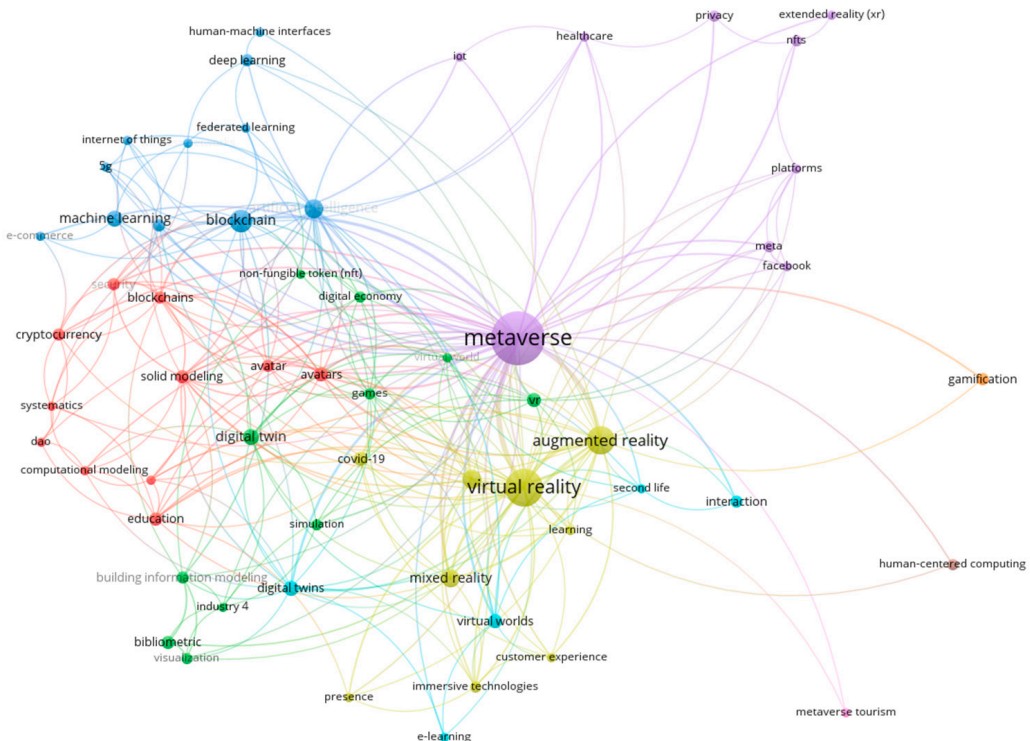

**Figure 7.** Author keyword clusters in the 491 metaverse-topic scientific publications in 2022 in the WoS database. Calculated with VOSviewer and based on keywords appearing at least three times in the WoS database.

Over the years, more authors have indicated directly that their publications were about the metaverse and technological devices, as well as features related to it. With this development, the studies no longer focused on Second Life, which no longer appear in the keywords, in 2022. In 2022, several clusters emerged indicating the main discussions around the publication keywords in the WoS. For example, the metaverse became associated with modeling, cryptocurrency, NFTs, and decentralized development. Virtual, augmented, and extended reality were connected to each other because they are digital environments. Blockchain became connected to artificial intelligence (AI) and machine learning, among other computational fields (Figure 8).

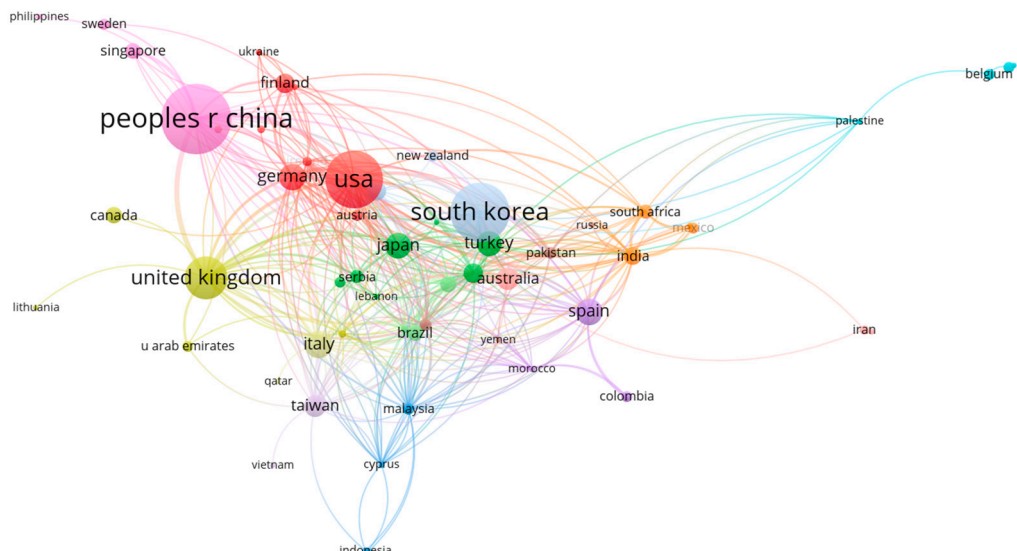

**Figure 8.** Geographical clustering of scientific metaverse-topic publications in the WoS database, 1995–2022. Calculated from the WoS database with VOSviewer.

Besides the aforementioned authors' metaverse keywords, we conducted content analysis on the publications of metaverse-topics between 1995–2022 in the WoS database. As indicated in Section 3, about material and methods, we established five main categories for the main foci of the articles: technologies, uses, societal influences, sustainability, and multi-topic reviews. Furthermore, we created more detailed sub-categories within these main categories (Table 3).

**Table 3.** Main and secondary contents in the scientific metaverse-topic publications the WoS database (%).

| Research Field | 1995–2021 | | | 2022 | | | All | | |
|---|---|---|---|---|---|---|---|---|---|
| | Main % | Total n | Total % | Main % | Total n | Total % | Main % | Total n | Total % |
| Uses | 48.1 | 98 | 62.0 | 39.6 | 161 | 48.3 | 42.4 | 259 | 52.7 |
| Technology | 26.6 | 55 | 34.8 | 31.8 | 126 | 37.8 | 30.1 | 181 | 36.9 |
| Societal Influence | 18.4 | 42 | 26.6 | 21.9 | 99 | 29.7 | 20.8 | 141 | 28.7 |
| Sustainability | 2.5 | 4 | 2.5 | 1.8 | 11 | 3.3 | 2.0 | 15 | 3.1 |
| Multitopic review | 1.9 | 4 | 2.5 | 4.5 | 15 | 4.5 | 3.7 | 19 | 3.9 |
| Unidentifiable | 2.5 | 4 | 2.5 | 0.3 | 1 | 0.3 | 1.0 | 5 | 1.0 |
| Total n | 100% | 158 | | 100% | 333 | | 100% | 491 | |

Until 2022, the most common main category was "uses" (48.1% of all publications, 76 publications), followed by "technology" (26.6%, 42), and "societal influence" (18.4%, 29). Very few of the publications (2.5%, 4) addressed sustainability as the main subject. The few remaining publications did not have any specified topic, but were broad reviews of the metaverse studies, or their topic other than the metaverse could not be defined, due to, for example, lacking an abstract and inaccessible full text version. More than two thirds of the publications (69.0%, 109) contained one clearly identifiable topic among these classes, the rest had also a secondary class assigned.

In more detail, the largest share of publications (13.9%, 22 publications) discussed metaverse-related software as the main topic. Equally, the most common theme was "education and learning" (13.9%, 22 articles), i.e., how various metaverse platforms can be used for educational purposes, and the differences between virtual and material environments, in achieving education-related impacts. A significant share (12.7%, 20) discussed how digitalization in work and business environments occurs, including the role of blockchains and NFTs in such processes. Of the publications, 17 (10.8%) concerned hardware. Arti-

cles regarding the societal impact of the metaverse discussed how the metaverse would cause changes in spaces (more abstractly) and places (more concretely), as well as its potential cultural and mental effects on people and their identities, emotions, security, and privacy [14,15].

In 2022, the most common main category was still "uses" (39.6% of all publications, main theme in 132 publications). A noteworthy growth occurred in publications addressing "technology" (31.8%, 106) and "societal influence" (21.0%, 73). An increasing number of scholars started considering the metaverse's effects on society, communities, and individuals. However, as aforementioned, the metaverse is far from being completed, so many of these publications were speculations about the future, instead of detailed empirical analyses. Proportionally, even fewer publications (1.8%, 6) addressed sustainability as their main topic, compared to earlier years. However, five articles had it as their secondary topic. The number and proportion of reviews (4.5%, 15) about the metaverse had increased because a corpus of studies existed that could be reviewed. One fourth of the publications (24.0%) contained a clearly identifiable secondary topic, often about applications.

The most common detailed content of the publications concerned metaverse-related software (14.7%, 49 publications). The next most common theme was education and learning. In 2022, it was work and business environments, which may also refer to changes in work environments in the (post-)COVID era, during which the practice of distance work increased, and possibly also the increasingly noted business opportunities related to metaverse topics. In addition, the number of publications referring to the metaverse's societal and other effects increased: for example, effects on communications and identities, emotions, and other individual-level aspects were handled more often. Hardware was also a topic in some publications (8.7%, 12).

*4.5. Sustainability as Topic in the Metaverse-Topic Publications*

In this article, we also analyzed international scientific peer-reviewed publications in which sustainability in the metaverse was the key topic. For this, we utilized the WoS database. Overall, only 11 scientific publications (2.6%) about the metaverse addressed sustainability as the main topic, and 4 as a secondary topic in the WoS. This small number of publications is rather curious because at the same time, many scientific and media articles foresaw that the metaverse would significantly affect global and local economies, social relations, and the physical environment ([1,7,9–17], see also Section 5 Discussion). A major global economic boost is expected from its implementation.

As evidenced by many publications about the metaverse, behind it are blockchain technologies, NFTs, and virtual social networks [1,24,25], which significantly affect global and local energy consumption and social relations. Another key aspect is access to the metaverse, in which digital divides matter, namely access to the Internet, as well as related uses and impacts. In 2021, less than two out of three of the global population (63%) were using the Internet, and in many less-developed countries this share was about a third of the population [26]. Therefore, much of the scientific discussion on the metaverse has so far focused on narrow technological or economic issues, and has not tied these topics to broader global and local sustainability issues or digital divides.

The topics in the publications that addressed the metaverse and sustainability are varied. Many dealt with very specific technological issues and mentioned sustainability only from a very narrow angle. However, we found exceptions that addressed sustainability more broadly. For example, Allam et al. (2022) argued that there were ethical, human, social, and cultural concerns in the metaverse's influence on the quality of human social interactions. The authors focused on the environmental, economic, and social goals of sustainability. The mapping of emerging products and services of the metaverse had the scope of reconstructing the quality of urban life, thus touching on the SDG 11 on sustainable cities and communities [27].

Also, Han (2022) referred to ethical issues in the metaverse, particularly AI ethics related to cultural and social sustainability. He referred to digital humans, AI avatars,

intelligent process automation, robots, cyborgs, and autonomous vehicles as examples that were emerging with AI and robot technology. Their ethics domains need to be addressed in multiple platforms, such as computing devices, intermediary platforms, and physical computing devices. One should use the metaverse to provide the educational basis for society's sustainability on ethical issues arising from the development of technology, thus indicating connections to the SDG 4 about quality education, despite not mentioning the SDGs directly [28].

Park and Kim (2022) also discussed the educational dimension of the metaverse. They directly referred to the SDGs in their article about sustainable learning (SDG 4). The metaverse would provide opportunities for gamification in learning processes, thereby increasing learners' motivation. According to the authors, the metaverse can provide learners equal educational opportunities by creating innovative educational environments [29].

Several articles on the metaverse and sustainability discussed the potential to enhance cities' technical design and construction processes, i.e., technologically assisted smart cities of the future. These publications had connections to the SDG 9 in innovation and infrastructure, as well as to SDG 11. For example, Liu et al. (2022) demonstrated that building-information modeling (BIM) brings about possibilities for reductions in carbon emissions, in the context of sustainable buildings [30]. Chen et al. (2022) presented the smart city as a mode to achieve sustainable construction. They also suggested deploying BIM to create smarter and more sustainable cities with their "Construction Metaverse" [31]. Aligned with that effort, Choi (2022) discussed how the metaverse's teleworking offices can reduce population pressure in megacities [32].

Energy efficiency was discussed in the article by Singh et al. (2022). They stated that current digital technologies would have a significant capability to realize sustainability in energy. In this, different contemporary and developing technologies, such as the IoT, AI, edge computing, blockchain, and big data, played an important role. They could be implemented to different stages of energy processes such as generation, distribution, transmission, smart grid, and energy trading. The main suggestion was to utilize big data for energy analytics, digital twins in smart grid modeling, virtual power plants with the metaverse, and green IoT for future enhancement on energy consumption [33]. Again, here, the SDG 7 about renewable energy is partly on the agenda.

Regarding social changes, Swiatek (2019) continued with the transformation of the traditional grey infrastructure of the old industry into the "invisible" infrastructure of Industry 4.0. The intelligent green-blue infrastructure would enlarge bio-productive lands, sustain biodiversity, and enforce regenerative abilities of coexisting ecosystems. Such a material–immaterial transformation will emerge as a digital "hypernature" that affects the material world [34]. In a similar vein, Neethirajan (2021) argued that deepfake technologies with enhanced social interactions could increase animal health, emotionality, sociality, and animal-human and animal-computer interactions. The claim was that the farming industry's productivity and sustainability would improve, loosely connected to SDG 15 about life on land [35].

More narrow themes are derived from metaverse and sustainability topics concerned, for example, the use of VR in product and packaging development, which would result in a more sustainable process requiring fewer resources and time than traditional procedures [36]. Similarly, the metaverse could be used for more sustainable cosmetic consumption and integrating it into life, health and beauty [37]. These themes are connected to the SDG 12 about responsible consumption.

Digitalization and immersive metaverse technologies would also impact tourism. On the one hand, it could reduce the volume of material tourism in concrete places. On the other hand, it would allow for expansion to new areas possible only (or mostly) through the metaverse, such as space travel [38]. Buhalis et al. (2022) expected that with the metaverse would emerge smart hospitality and new infrastructures that will bring more sustainability. These would be disruptive innovations in the entire hospitality ecosystem and marketing-driven hospitality [39]. These themes have connections to the SDGs 9 and 12.

The metaverse could also be used to enhance the contemporary material world by utilizing digital twins to provide better estimations of development. Kwoo et al. (2022) suggested that digital twins could be used to enhance high-fidelity disaster models and real-time observational environmental data, with distributed computing schemes. It would substantially decrease the disaster prediction time [40].

## 5. Discussion

The metaverse is still in an emerging phase. Therefore, much of the scientific analysis of it in the 2010s was future speculation instead of measurement of its effects. Some online games and virtual reality environments were studied also from the viewpoint of impacts, Second Life being the most commonly analyzed. In the 2020s, scientific research on the metaverse has started to boom. Still, much of the research is about small details of the metaverse and technological aspects regarding its potential uses (see Section 6 Conclusions below). The uses have not been addressed much because the metaverse is still far from its completion.

The metaverse will become completed in the future—this is the prediction of many scholars and expressions in the media. The general and specialized media have estimated that the metaverse will have a huge economic impact, increasing the global economy by several percentage points [2,9,10]. The metaverse is expected to significantly affect social communication and networks, perhaps even more than social media has done so far. Its impact on education and learning can be significant. There are many opportunities, but also open issues. The metaverse contributes substantially to global and local sustainability. This has not yet been addressed in depth in the scientific publications. Is the metaverse supporting or constraining to achieve the SDGs that all countries in the world are committed to achieve by 2030? How would metaverse development to SDGs and environmental, social, and economic sustainability be better achieved? Can machine learning with deep learning, on big data through the metaverse, provide assets on how to address sustainability?

Sustainability needs to be addressed more in the development of the metaverse. It is also an ethical requirement for metaverse development. If sustainability does not become the key concern and goal, there will be no metaverse in the long term. A more holistic and integrated approach to sustainability is necessary in scientific studies on the metaverse. That would be a kind of impact assessment—how this or that metaverse topic is connected to sustainable development, and which aspects of sustainability, and whether there are better ways to achieve short- and long-term sustainability through the metaverse. This does not involve withdrawing from the goal of the metaverse contributing substantially to economic growth in the 2030s. However, such growth needs to enhance sustainability in responsible ways.

With regards to environmental sustainability, the metaverse could reduce material consumption and the use of natural resources, if consumption connected to economic and symbolic values could occur digitally, preserving natural resources and the environment. Turning many aspects into digital twins should not require people to turn to the digital world to escape the challenges of the material world, or double the sustainability challenges, because these challenges would occur both in the physical and digital worlds. Moreover, one needs to address the energy and natural resources required in the construction of the metaverse and its devices, as well as in its operationalization. Research could address more precisely the environmental impacts of the metaverse while designing, constructing, and using it.

Regarding social sustainability, there is a multitude of topics that need to be researched, starting from the digital divides constraining the access, to the metaverse, online monitoring, privacy, and detection in the metaverse, and the meaning of life with and without the metaverse. Here we mention two interconnected themes, namely language and culture. English is the predominant language in which the scientific discussion about the metaverse takes place, as it is the most common language in international peer-reviewed publications. It is important to know what is discussed scientifically about the metaverse in English,

in this emerging and expanding phase of related research. Obviously, the metaverse is also discussed scientifically in languages other than English. In addition, the majority of discussions take place outside the scientific arena and in a variety of languages. A whole new vocabulary is emerging, initially in English and then possibly with or without translation immersed into other languages. When the metaverses become in the future widely used platforms, many languages will be used there to facilitate social networking and economic activities. Therefore, it would be important that scientific research on the metaverse and especially about its uses would be conducted in many languages. It is important to study the uses of the metaverse in different linguistic and cultural environments. This will show evidence on how much the metaverse(s) can maintain or support diversity or whether as a technological-digital platform it can reduce the cultural diversity of places around the world, and whether the metaverse itself can become culturally diverse in visual and social dimensions. A connected theme to be studied is the cultural diversity within the metaverse—will there be digital twins of existing cultures, will it be a melting pot or something in which cultures connected local material environments interact with those in the digital realms, thus developing hybrid material-digital cultures.

Economic sustainability of the metaverse is a key topic whether the metaverse will ever be and in which forms. Technology developers and enterprises have already invested huge amounts of money and time to develop the metaverse. However, they are not there yet, so we are not there yet. At the same time, in the physical material world are several sustainability challenges. Many SDGs could have been tackled with the money and time invested into the metaverse. Loosely connected to the metaverse, in the recent years we have evidenced rapid and unsustainable changes in cryptocurrencies and NFTs that are expected to be important in the metaverse. It is crucial that the ambitions of economic growth in developing the metaverse will not create economic unsustainability. This is also a topic to be studied, i.e., how the economic system in the metaverse is organized, how economic profits are generated and how they are shared there. The metaverse also provides an opportunity for novel innovation processes in which novel knowledge is created and designed from ideas to innovations in human-machine interaction [41]. This provides opportunities for better matching of sustainable and responsible development, along broader value-added network [42]. Overall, economic sustainability of the metaverse is in many ways connected to social and environmental sustainability.

## 6. Conclusions

The number of scientific publications about the metaverse started to increase in the 2020s. In 2022, the volume of scientific metaverse-topic publications doubled. There are over 300 peer-reviewed scientific publications in international journals and conference proceedings indexed in the WoS, and over 1100 publications in Scopus. In the near future, thousands of articles on the metaverse may be published annually.

The key countries and organizations behind these metaverse-related publications are in general those that are highly ranked among those publishing internationally. Scholars and institutions from the United States, China, the United Kingdom, and South Korea are the countries most frequently behind the publications on the metaverse. The growth of publications of metaverse research from China is advancing very fast. Selected universities and research institutes from China appear among those in which related publication activity is the largest but that is fast also in some universities in Singapore, Japan, and South Korea. A specific geographical concentration of metaverse-related research has emerged in Asia, though connections between these sites is not dense, at least so far. At the same time, the number of countries with organizations and scholars producing scientific articles about the metaverse is expanding quickly. Still many scholars and institutions from Africa do not have any scientific publications about the metaverse and rather few originate from the Central Asia either.

The metaverse-related publications' key research topics and areas were about the variety of its uses and related technologies. Much of the scientific research is about computer

science, engineering, and other topics that are connected to the metaverse's technological characteristics. In the end, it develops alongside technological platforms, devices, and connections. A few research topics and area clusters have emerged. These include, for example, themes on virtual and augmented reality, those on blockchain, cryptocurrencies and NFTs, those on modeling and machine learning, and those loosely related to varied aspects of decentralization.

No one knows how the metaverse will be in the 2030s, and what kind of metaverses will be present then. Nevertheless, many scholars argue that the metaverse will have a significant impact on the global economy and modify people's social relations, of which there is already evidence with the emergence of social media through the Internet. Very little of the metaverse-related research has so far addressed sustainability, despite the presumed major impacts of the metaverse on environmental, social, and economic sustainability. To support the SDGs could be a useful framework on which to analyze and develop the metaverse now and in the future. Digital divides matter, so for a broader and necessary social sustainability of the metaverse, everyone should have the opportunity to access it regardless of their location or social or economic position.

**Author Contributions:** Conceptualization, J.S.J.; methodology, J.S.J.; software, J.S.J., C.K. and J.J.; validation, J.S.J., C.K. and J.J.; formal analysis, J.S.J., C.K. and J.J.; investigation, J.S.J.; resources, J.S.J.; data curation, J.S.J., C.K. and J.J.; writing—original draft preparation, J.S.J., C.K. and J.J.; writing—review and editing, J.S.J.; visualization, J.S.J., C.K. and J.J.; supervision, J.S.J.; project administration, J.S.J.; funding acquisition, J.S.J. All authors have read and agreed to the published version of the manuscript.

**Funding:** The research was funded by the University of Turku.

**Institutional Review Board Statement:** Not applicable.

**Informed Consent Statement:** Not applicable.

**Data Availability Statement:** Not applicable.

**Conflicts of Interest:** The authors declare no conflict of interest. The funders had no role in the design of the study; in the collection, analyses, or interpretation of data; in the writing of the manuscript, or in the decision to publish the results.

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
