# Peer review of "Metaverse and Sustainability: Systematic Review of Scientific Publications until 2022 and Beyond"

_sustainability, doi:10.3390/su15010346_

Round 1
Reviewer 1 Report
My main concerns are:
The abstract presents the contributions of the article but do not indicate the main findings of the survey.
The authors "focused on Metaverse and Sustainability”, however, this decision is not clear. They must include a better explanation of the contribution to the introduction section, as well as explain about online monitoring, privacy, detection, ... . To illustrate the contribution, I suggest adding a figure which shows technological advances throughout the last years.
The survey paper details the most used technologies. The discussion of these mechanics is dramatically relevant to this work, and an adequate introduction of each of them is more than appropriate. For instance, “Machine and Deep Learning” has widely been developed in several research lines. I propose improving each section in order to introduce the technologies and techniques in a better way.
I expected to read the leasson learned, research opportunities, new proposals, correlate it with other current Technologies, such as: IoT (communications, networks, Cloud, …), in terms of latency I guess that this field is quite sensitive to the delays required to process data, which should call for new investigations around the tradeoff between learning cost and performance (e.g. Deep Learning is costly, yet attains good predictive scores… should we opt for weak learners over good features? Or complex learners over raw data? Or a mixture of both of them, e.g. learned features off-line + weak learners on-line? Should data be sent to the cloud? Be preprocessed at the edge?). This issue is also very trendy at the communications level.
Discussion should be placed along Opportunities and Open Issues in a single section (e.g., “Discussion, Opportunities and Open Issues”), and one open issue should be the lack of a thorough reference model for future metaverse developments, so that innovations in these platforms can be easily identified, exchanged, analyzed, exported, etc. And then the reference model in a new section. In this way the contribution to the field would be very justified.
Open issues: the current content is rather unclear in regards to the specific challenges that remain unsolved. Instead, it merely describes several works that are allegedly aligned with a certain open issue, but the open issue itself is not described concisely. I would also suggest including aspects related to data such as ethics.
The reference model is one of the major contributions of the paper, and, thus, the authors could enrich it with further information, specifications and functionalities that should be agreed prior to the design and implementation of each phase, examples, performance measures, …?
Author Response
Reviewer 1
Response: Thank you for your very valuable comments and suggestions. We have taken them into account as much as possible as evidenced by our responses and in the revised manuscript.
R1.1 The abstract presents the contributions of the article but do not indicate the main findings of the survey.
Response: The abstract has been now modified indicate the main findings
R1.2 The authors "focused on Metaverse and Sustainability”, however, this decision is not clear. They must include a better explanation of the contribution to the introduction section, as well as explain about online monitoring, privacy, detection, ... . To illustrate the contribution, I suggest adding a figure which shows technological advances throughout the last years.
Response: The contribution is now clarified. Online monitoring, privacy and detection are added to the discussion about important research topic about the metaverse. The topic of the article is not the development of the metaverse but how the metaverse has been studied in scientific publications. Unfortunately, the word limits do not offer possibility to integrate a timeline of the metaverse development.
R1.3 The survey paper details the most used technologies. The discussion of these mechanics is dramatically relevant to this work, and an adequate introduction of each of them is more than appropriate. For instance, “Machine and Deep Learning” has widely been developed in several research lines. I propose improving each section in order to introduce the technologies and techniques in a better way.
Response: We have now clarified some of these terms, however, the focus of the articles is not in these terms but how the metaverse has been studied in scientific publications.
R1.4 I expected to read the leasson learned, research opportunities, new proposals, correlate it with other current Technologies, such as: IoT (communications, networks, Cloud, …), in terms of latency I guess that this field is quite sensitive to the delays required to process data, which should call for new investigations around the tradeoff between learning cost and performance (e.g. Deep Learning is costly, yet attains good predictive scores… should we opt for weak learners over good features? Or complex learners over raw data? Or a mixture of both of them, e.g. learned features off-line + weak learners on-line? Should data be sent to the cloud? Be preprocessed at the edge?). This issue is also very trendy at the communications level.
Response: The scope of the article is to indicate what has been studied regarding the metaverse. Some of these themes have now been added to the future research avenues.
R1.5 Discussion should be placed along Opportunities and Open Issues in a single section (e.g., “Discussion, Opportunities and Open Issues”), and one open issue should be the lack of a thorough reference model for future metaverse developments, so that innovations in these platforms can be easily identified, exchanged, analyzed, exported, etc. And then the reference model in a new section. In this way the contribution to the field would be very justified.
Response: We follow the guidelines of the journal and have the title of Discussion but also mention now there also issues on opportunities and open issues and provide an example of a metaverse-based innovation platform.
R1.5 Open issues: the current content is rather unclear in regards to the specific challenges that remain unsolved. Instead, it merely describes several works that are allegedly aligned with a certain open issue, but the open issue itself is not described concisely. I would also suggest including aspects related to data such as ethics.
Response: We have now clarified more the contribution of sustainability-related articles on the metaverse. Ethical issues have now been addressed in material and methods section.
R1.6 The reference model is one of the major contributions of the paper, and, thus, the authors could enrich it with further information, specifications and functionalities that should be agreed prior to the design and implementation of each phase, examples, performance measures, …?
Response: The material and methods section has now been revised and showed in detail about the data collection and analysis.
Reviewer 2 Report
The text is very interesting and current. Reading the article, brought several academic insights to the reviewer who is an expert on the subject.
I hope that the authors find in the revisions a way to improve the text and transform it into an academic reference for future generations. The text is being written at a crucial moment in the development of the Metaverse and new technological platforms, which can indeed contribute to a more aware generation of people.
Below are the reviewer's contributions to the authors:
Point 1. Summary. Lines 13 to 19. Why was the search done solely on the Web of Science database? There are other good databases, such as those indexed by Scopus. Please justify.
When the work is a reviewed, the authors can use the PRISMA systematic methodology in at least two or three worldwide databases.
Point 1.1: review whether the summary covers the purpose of the study, the relevance of the topic, the methodology used, and the results.
Point 2. Figure 1. Bring the original source in color and in high resolution. Modifications can be made and contextualized in the text. Authors can choose to make an original.
Point 3. Lines 86 to 89. The research needs to be systematized by the authors to have a real vision of what is happening.
Examples: How to apply PRISMA in academic practice "Azevedo Guedes et al. (2018)" or PRISMA - TRANSPARENT REPORTING of SYSTEMATIC REVIEWS and META-ANALYSES (https://prisma-statement.org/).
Point 4. Lines 104 and 105. Seek to systematize the results found so that they are scientifically more easily visualized through an illustrative figure and with the inclusion and exclusion criteria that PRISMA provides.
The introduction needs to be improved to provide an overview of the subject matter (contextualization), contemplate the relevance of the subject (justification), present why it was elaborated (objective), present the research problem or research problems, mention previous works that address the topic in question and generically address what will be studied in the rest of the article (text structure).
Point 5. Lines 119 to 122. How will carbon emissions be reduced? Bring the scientific basis to the statements. This is a good talking point.
Could it be that more and more data centers around the world would not substantially increase carbon emissions? It lacked academic substance.
Point 6. Lines 125 to 127. Bring authors or companies that corroborate this view. Cite in the text and reference.
Point 7. Lines 147 to 150. This is an important point of the work.
How to generate less poverty, respect for the environment, and more income opportunities with Metaverse? I observe that, in essence, it is still expensive and inaccessible for developing countries (mainly third-world ones).
Point 8. Line 178 to 183. Google Scholar does not have the academic quality to be considered, however, SCOPUS does.
The solution is to assemble PRISMA only from WOS and better justify the absence of SCOPUS, which has many relevant articles on the subject, or even redo the search.
Remove the following excerpt: "the WoS is the best database for that purpose."
Point 9. Lines 200 to 209. By the criteria adopted, the precursor of this movement in Information Technology and the Internet was excluded from the research and can be considered a scientific imprecision.
The " Second Life" game that the authors cite later was a Metaverse in essence and the initial release took place in 2003 by the company Linden Lab. Many scientists debated the same at the time.
Point 10. Lines 229 to 235. What other games could be considered metaverses and were left out of the research by the adopted criteria? To quote.
Point 11. Lines 254 to 257. The quoted figure is outside the MDPI standard and with a low-quality image.
Point 12. Lines 270 to 275. Mention the main scientific journals - which contributed the most with articles - that discussed the Metaverse theme and which were considered by the authors in the research.
Point 13. Lines 299 to 308. The research bias could have brought countries only by the database used, after the insertion of PRISMA justified or expanded the research to other bases.
Point 14. Figures 5, 6 and 7 are outside MDPI publication standards with low visual quality (resolution). Below or above each figure insert a text between them referencing their importance for the article, as well as the reflections that the figures bring to the research.
Point 15. Lines 354 to 359. Based on the literature, assemble an explanatory table of the importance of modeling, cryptocurrencies and NFTs to the Metaverse.
Why topics such as blockchain, decentralized development, DAOs, digital twins, data privacy, among others, did not have the same importance?
Bring the view of the authors justified by the literature.
Point 16. The analysis of the works of literature does not support traditional sustainability (environmental, social, and economic) as a crucial point of the development of the Metaverses.
Authors should focus on the relevant points of the United Nations 2030 Agenda and the Sustainable Development Goals that make more sense to the results found in the literature.
Perceptibly, sustainability is being biased by researchers, against more relevant ones, such as quality education or economic growth. How can authors bring insights into this new world with the SDGs?
The comments are valid for reflection on the item "4.5. Sustainability Topic in the metaverse-topic Publications " which I find very interesting but needs to be revised in light of previous comments.
Point 17. 5. Discussion. The text lacked a specific chapter to bring the statistical results found even without discussing them in depth. The reviewer missed tables, figures, and graphs to summarize the data. Normally, statistical tables and figures can be used to reduce the amount of text, display numerical values, explain variables, or present answers to questions, in addition to providing greater visual impact.
Introduce a chapter on the Analysis of the Results, before the Conclusions. The analysis of the results brings a synthesis of the results obtained in light of the theoretical framework presented in the bibliographic review. In this sense, it interprets, evaluates, and criticizes the results obtained, considering the existing knowledge on the subject, including references from other works to support or compare the results. It should also address the importance of the results, their limitations, implications, and possible applications. As far as possible, some possible developments for future work can also be suggested.
The tables, figures, and graphs also used above can serve to summarize the analysis.
Finally, include a chapter of conclusions and improve the final text. They present in a summarized and objective way the relevant elements addressed during the argumentation and can be considered an interpretative summary of the observations and experiments. In this sense, they must be analytical, and interpretive, and include explanatory arguments.
In addition, they should highlight how each of the objectives was achieved. The questions asked by the authors at the beginning of the text (lines 93 to 101) were not objectively answered. Only the elements considered in the text should be addressed, that is, new elements should not be presented and discussed.
Author Response
Reviewer 2
The text is very interesting and current. Reading the article, brought several academic insights to the reviewer who is an expert on the subject. I hope that the authors find in the revisions a way to improve the text and transform it into an academic reference for future generations. The text is being written at a crucial moment in the development of the Metaverse and new technological platforms, which can indeed contribute to a more aware generation of people. Below are the reviewer's contributions to the authors:
Response: Thank you for your very valuable comments and suggestions. We have taken them into account as much as possible as evidenced by our responses and in the revised manuscript.
R2.1 Point 1. Summary. Lines 13 to 19. Why was the search done solely on the Web of Science database? There are other good databases, such as those indexed by Scopus. Please justify.
When the work is a reviewed, the authors can use the PRISMA systematic methodology in at least two or three worldwide databases
Response: Thank you for the suggestion. The WoS is been used as a more narrow and sharp data on international peer-reviewed scientific publications. We have now added also Scopus from broader analysis of the metaverse-related publication numbers, themes, countries of origin, institutions, etc. PRISMA is now applied and indicated with a new figure.
Point 1.1: review whether the summary covers the purpose of the study, the relevance of the topic, the methodology used, and the results.
Response: Abstract and conclusions have now been revised accordingly.
R2.2 Point 2. Figure 1. Bring the original source in color and in high resolution. Modifications can be made and contextualized in the text. Authors can choose to make an original.
Response: Figure has been redrawn. The quality in Word-file is not the same than in the jpg.-version.
R2.3 Point 3. Lines 86 to 89. The research needs to be systematized by the authors to have a real vision of what is happening.
Examples: How to apply PRISMA in academic practice "Azevedo Guedes et al. (2018)" or PRISMA - TRANSPARENT REPORTING of SYSTEMATIC REVIEWS and META-ANALYSES (https://prisma-statement.org/).
Response: PRISMA has now been used.
R2.4 Point 4. Lines 104 and 105. Seek to systematize the results found so that they are scientifically more easily visualized through an illustrative figure and with the inclusion and exclusion criteria that PRISMA provides.
The introduction needs to be improved to provide an overview of the subject matter (contextualization), contemplate the relevance of the subject (justification), present why it was elaborated (objective), present the research problem or research problems, mention previous works that address the topic in question and generically address what will be studied in the rest of the article (text structure).
Response: Introduction has been now revised as well as conclusions.
R2.5 Point 5. Lines 119 to 122. How will carbon emissions be reduced? Bring the scientific basis to the statements. This is a good talking point.
Could it be that more and more data centers around the world would not substantially increase carbon emissions? It lacked academic substance.
Response: Agree, it depends how all this is organized. Phrases clarified and references added.
R2.6 Point 6. Lines 125 to 127. Bring authors or companies that corroborate this view. Cite in the text and reference.
Response: Phrases clarified and references added.
R2.7 Point 7. Lines 147 to 150. This is an important point of the work.
How to generate less poverty, respect for the environment, and more income opportunities with Metaverse? I observe that, in essence, it is still expensive and inaccessible for developing countries (mainly third-world ones).
Response: Issues of digital divides discussed here and in the Discussion more clearly and many sustainability issues risen in the conclusions.
R2.8 Point 8. Line 178 to 183. Google Scholar does not have the academic quality to be considered, however, SCOPUS does.
The solution is to assemble PRISMA only from WOS and better justify the absence of SCOPUS, which has many relevant articles on the subject, or even redo the search.
Remove the following excerpt: "the WoS is the best database for that purpose."
Response: PRISMA applied to WoS and Scopus also added as the data material. Excerpt modified accordingly.
R2.9 Point 9. Lines 200 to 209. By the criteria adopted, the precursor of this movement in Information Technology and the Internet was excluded from the research and can be considered a scientific imprecision.
The " Second Life" game that the authors cite later was a Metaverse in essence and the initial release took place in 2003 by the company Linden Lab. Many scientists debated the same at the time.
Response: A few points on Second Life mentioned more and more clearly. In the most recent metaverse-related research the scholars do not anymore write about Second Life.
R2.10 Point 10. Lines 229 to 235. What other games could be considered metaverses and were left out of the research by the adopted criteria? To quote.
Response: We did not study the games but referred to the games that scholars had studied.
R2.11 Point 11. Lines 254 to 257. The quoted figure is outside the MDPI standard and with a low-quality image.
Response: The quality in Word-file is not the same than in the jpg.-version.
R2.12 Point 12. Lines 270 to 275. Mention the main scientific journals - which contributed the most with articles - that discussed the Metaverse theme and which were considered by the authors in the research.
Response: The list of the main journals added.
R2.13 Point 13. Lines 299 to 308. The research bias could have brought countries only by the database used, after the insertion of PRISMA justified or expanded the research to other bases.
Response: The provenience of the articles has been double checked. Similar analysis has now been conducted on the Scopus.
R2.14 Point 14. Figures 5, 6 and 7 are outside MDPI publication standards with low visual quality (resolution). Below or above each figure insert a text between them referencing their importance for the article, as well as the reflections that the figures bring to the research.
Response: The quality in Word-file is not the same than in the jpg.-version.
R2.15 Point 15. Lines 354 to 359. Based on the literature, assemble an explanatory table of the importance of modeling, cryptocurrencies and NFTs to the Metaverse. Why topics such as blockchain, decentralized development, DAOs, digital twins, data privacy, among others, did not have the same importance? Bring the view of the authors justified by the literature.
Response: Why far more authors addressed modeling, NFTs, etc., we do not know. The focus of analysis was on showing which topics were addressed more and which less. The judging of what should have been addressed falls out of the scope of the article. Decentralization is now added after the second look of the material. However, in the discussion have been brought some of these issues present.
R2.16 Point 16. The analysis of the works of literature does not support traditional sustainability (environmental, social, and economic) as a crucial point of the development of the Metaverses. Authors should focus on the relevant points of the United Nations 2030 Agenda and the Sustainable Development Goals that make more sense to the results found in the literature. Perceptibly, sustainability is being biased by researchers, against more relevant ones, such as quality education or economic growth. How can authors bring insights into this new world with the SDGs? The comments are valid for reflection on the item "4.5. Sustainability Topic in the metaverse-topic Publications " which I find very interesting but needs to be revised in light of previous comments.
Response: The connections to the SDGs has now been added. In Discussion is discussed about now economic, social and environmental sustainability at length.
R2.17 Point 17. 5. Discussion. The text lacked a specfic chapter to bring the statistical results found even without discussing them in depth. The reviewer missed tables, figures, and graphs to summarize the data. Normally, statistical tables and figures can be used to reduce the amount of text, display numerical values, explain variables, or present answers to questions, in addition to providing greater visual impact.
Response: The Table 3 indicates the main results as well as figures in the section 4. The main results are now written in the new chapter 6 Conclusions.
R2.18 Introduce a chapter on the Analysis of the Results, before the Conclusions. The analysis of the results brings a synthesis of the results obtained in light of the theoretical framework presented in the bibliographic review. In this sense, it interprets, evaluates, and criticizes the results obtained, considering the existing knowledge on the subject, including references from other works to support or compare the results. It should also address the importance of the results, their limitations, implications, and possible applications. As far as possible, some possible developments for future work can also be suggested.
Response: The sections 5 Discussion and 6 Conclusions (with results of research questions) are now separated.
R2.19 The tables, figures, and graphs also used above can serve to summarize the analysis.
Response: The main detailed empirical results are indeed in tables and figures.
R2.20 Finally, include a chapter of conclusions and improve the final text. They present in a summarized and objective way the relevant elements addressed during the argumentation and can be considered an interpretative summary of the observations and experiments. In this sense, they must be analytical, and interpretive, and include explanatory arguments.
Response: New Section 6 Conclusions is now added.
R2.21 In addition, they should highlight how each of the objectives was achieved. The questions asked by the authors at the beginning of the text (lines 93 to 101) were not objectively answered. Only the elements considered in the text should be addressed, that is, new elements should not be presented and discussed.
Response: These are now precisely answered in the new Section 6 Conclusions.
Reviewer 3 Report
Dear author(s),
Thanks for your submission and for your interesting contribution. Despite the paper is well developed I suggest some major revision allowing you a better refinement of your research.
In general, your paper suffers from the problems of the mostly bibliometric/review papers. You are offering some snapshot of the literature and a huge amount of data and tables, without offering a deep analysis of the available literature.
Also, the method is not solid, while there is a need to guarantee the reproducibility of your results. Please find my comments in detail below with some already published papers that could give you inspiration on how to re-frame your study.
Here are my comments:
1. Can you please clearly specify the rationale of the paper? Why is it needed? It is not clear from the introduction. Please specify the boundaries and the RQ of your study.
2. The same applies to the implications of your study. Can you please better stratify the results of your literature analysis?
3. The methodological steps need to be highly revised. The methodology is not clear, why bibliometric is useful in this case? What are the boundaries of your study? How and why did you excluded/included papers? I even suggest reporting the full list of papers as supplementary material.
You can look at the following paper for guidance on the three aforementioned points:
Turzo, T., Marzi, G., Favino, C., & Terzani, S. (2022). Non-financial reporting research and practice: Lessons from the last decade. Journal of Cleaner Production, 131154.
4. I suggest a better exploration of your research questions. For example, when exploring the future research avenues much more can be done in guiding the scholars. In the current form, the paper is completely missing future avenues.
You can look here:
Marzi, G., Ciampi, F., Dalli, D., & Dabic, M. (2020). New product development during the last ten years: The ongoing debate and future avenues. IEEE Transactions on Engineering Management, 68(1), 330-344.
Good luck with your review.
Author Response
Reviewer 3
Thanks for your submission and for your interesting contribution. Despite the paper is well developed I suggest some major revision allowing you a better refinement of your research. In general, your paper suffers from the problems of the mostly bibliometric/review papers. You are offering some snapshot of the literature and a huge amount of data and tables, without offering a deep analysis of the available literature. Also, the method is not solid, while there is a need to guarantee the reproducibility of your results. Please find my comments in detail below with some already published papers that could give you inspiration on how to re-frame your study. Here are my comments:
Response: Thank you for your very valuable comments and suggestions. We have taken them into account as much as possible as evidence by our responses and in the revised manuscript.
R3.1. Can you please clearly specify the rationale of the paper? Why is it needed? It is not clear from the introduction. Please specify the boundaries and the RQ of your study.
Response: Research questions are now clearly presented in the introduction as well as the aim of the article.
R3.2. The same applies to the implications of your study. Can you please better stratify the results of your literature analysis?
Response: A new section 6 Conclusions has been added indicating clear answers to each research questions.
R3.3. The methodological steps need to be highly revised. The methodology is not clear, why bibliometric is useful in this case? What are the boundaries of your study? How and why did you excluded/included papers? I even suggest reporting the full list of papers as supplementary material.
You can look at the following paper for guidance on the three aforementioned points:
Turzo, T., Marzi, G., Favino, C., & Terzani, S. (2022). Non-financial reporting research and practice: Lessons from the last decade. Journal of Cleaner Production, 131154.
Response: The materials and methods section has now been fully revised and methodological steps have now been clarified, and PRISMA used, and a reference to Turzo added.
R3.4. I suggest a better exploration of your research questions. For example, when exploring the future research avenues much more can be done in guiding the scholars. In the current form, the paper is completely missing future avenues.
You can look here:
Marzi, G., Ciampi, F., Dalli, D., & Dabic, M. (2020). New product development during the last ten years: The ongoing debate and future avenues. IEEE Transactions on Engineering Management, 68(1), 330-344.
Response: These are now being discussed in fully revised section 5 Discussion. The avenues regard the three dimensions of sustainability.
Round 2
Reviewer 2 Report
The text is very interesting and current. Reading the article, brought several academic insights to the reviewer who is an expert on the subject.
I hope that the authors find in the revisions a way to improve the text and transform it into an academic reference for future generations. The text is being written at a crucial moment in the development of the Metaverse and new technological platforms, which can indeed contribute to a more aware generation of people.
First, the reviewer thanks the authors for presenting a text with higher quality and rigorous bibliographical research.
The text also had a significant improvement in its content regarding the SDGs. It may have gone unnoticed by the authors, but figure 4 referring to PRISMA needs to be adjusted to demonstrate the process carried out in both databases (Scopus and WoS), even if unifying the searches in different databases. The inclusion and exclusion criteria are: the selection filter, as well as the keywords, and the time period (interval between dates) of the search performed.
Finally, if the authors could make the excel file with the mentioned articles available for review by the reviewers of the article, that would be great.
Author Response
Thank you for your comments. The Figure 4 flow char has now been revised as your suggested.
Reviewer 3 Report
Dear authors,
you successfully addressed all of my comments. I, therefore, suggest acceptance of the paper.
Author Response
Thank you for your comments.
Round 3
Reviewer 2 Report
The article gained quality and technicaly consistence.